# The Prognostic Value of Lymph Node Involvement after Neoadjuvant Chemotherapy Is Different among Breast Cancer Subtypes

**DOI:** 10.3390/cancers13020171

**Published:** 2021-01-06

**Authors:** Lucie Laot, Enora Laas, Noemie Girard, Elise Dumas, Eric Daoud, Beatriz Grandal, Jean-Yves Pierga, Florence Coussy, Youlia Kirova, Elsy El-Alam, Guillaume Bataillon, Marick Lae, Florence Llouquet, Fabien Reyal, Anne-Sophie Hamy

**Affiliations:** 1Department of Surgical Oncology, Institute Curie, University Paris, 75005 Paris, France; lucie.laot@aphp.fr (L.L.); enora.laas@curie.fr (E.L.); noemie.girard@curie.fr (N.G.); florence.llouquet@curie.fr (F.L.); 2Residual Tumor & Response to Treatment Laboratory, RT2Lab, Translational Research Department, INSERM, U932 Immunity and Cancer, INSERM, University Paris, 75005 Paris, France; elise.dumas@curie.fr (E.D.); eric.daoud@curie.fr (E.D.); beatriz.grandalrejo@curie.fr (B.G.); anne-sophie.hamy-petit@curie.fr (A.-S.H.); 3Department of Medical Oncology, Institute Curie, University Paris, 75005 Paris, France; jean-yves.pierga@curie.fr (J.-Y.P.); florence.coussy@curie.fr (F.C.); 4Department of Radiotherapy, Institute Curie, University Paris, 75005 Paris, France; youlia.kirova@curie.fr; 5Department of Tumor Biology, Institute Curie, University Paris, 75005 Paris, France; elsy.elalam@curie.fr (E.E.-A.); guillaume.bataillon@curie.fr (G.B.); 6Henri Becquerel Cancer Center, Department of Pathology, INSERM U1245, UniRouen Normandy University, 76130 Rouen, France; marick.lae@chb.unicancer.fr

**Keywords:** breast cancer, neoadjuvant chemotherapy, prognostic, residual axillary disease, nodal involvement, number of positive nodes

## Abstract

**Simple Summary:**

Little is known about whether residual axillary disease after neoadjuvant chemotherapy carries a different prognostic value by breast cancer subtype. We retrospectively evaluated the axillary involvement (0, 1 to 3 positive nodes, ≥4 positive nodes) on surgical specimens from a cohort of 1197 patients treated with neoadjuvant chemotherapy, and analyzed its association with survival outcomes. Relapse free survival was significantly associated with the number of positive nodes, but this effect was different by breast cancer subtype (Pinteraction = 0.004). High risk patients were those with 4 or more nodes involved in the luminal subgroup, whereas patients with 1 node or more involved had a decreased prognosis in triple negative and *HER2* positive breast cancer subgroups. The prognostic value of residual axillary disease should be interpreted according to breast cancer subtype to accurately stratify patients with a high risk of recurrence after neoadjuvant chemotherapy who should be offered second line therapies.

**Abstract:**

Introduction: The three different breast cancer subtypes (Luminal, *HER2-*positive, and triple negative (TNBCs) display different natural history and sensitivity to treatment, but little is known about whether residual axillary disease after neoadjuvant chemotherapy (NAC) carries a different prognostic value by BC subtype. Methods: We retrospectively evaluated the axillary involvement (0, 1 to 3 positive nodes, ≥4 positive nodes) on surgical specimens from a cohort of T1-T3NxM0 BC patients treated with NAC between 2002 and 2012. We analyzed the association between nodal involvement (ypN) binned into three classes (0; 1 to 3; 4 or more), relapse-free survival (RFS) and overall survival (OS) among the global population, and according to BC subtypes. Results: 1197 patients were included in the analysis (luminal (*n* = 526, 43.9%), TNBCs (*n* = 376, 31.4%), *HER2*-positive BCs (*n* = 295, 24.6%)). After a median follow-up of 110.5 months, ypN was significantly associated with RFS, but this effect was different by BC subtype (*P*_interaction_ = 0.004), and this effect was nonlinear. In the luminal subgroup, RFS was impaired in patients with 4 or more nodes involved (HR 2.8; 95% CI [1.93; 4.06], *p* < 0.001) when compared with ypN0, while it was not in patients with 1 to 3 nodes (HR = 1.24, 95% CI = [0.86; 1.79]). In patients with TNBC, both 1-3N+ and ≥4 N+ classes were associated with a decreased RFS (HR = 3.19, 95% CI = [2.05; 4.98] and HR = 4.83, 95% CI = [3.06; 7.63], respectively *versus* ypN0, *p* < 0.001). Similar decreased prognosis were observed among patients with *HER2*-positive BC (1-3N +: HR = 2.7, 95% CI = [1.64; 4.43] and ≥4 N +: HR = 2.69, 95% CI = [1.24; 5.8] respectively, *p* = 0.003). Conclusion: The prognostic value of residual axillary disease should be considered differently in the 3 BC subtypes to accurately stratify patients with a high risk of recurrence after NAC who should be offered second line therapies.

## 1. Introduction

Neoadjuvant chemotherapy (NAC) has been for decades the cornerstone of treatment strategy for locally advanced breast cancers (BC) (T3-T4), and tumors not accessible to conservative treatment. Since the publication of the CREATE-X [1] and the KATHERINE trial [2], new post-neoadjuvant treatment options have emerged in triple negative (TNBCs) and *HER2*-positive BC. Beyond the increase of breast conservative surgery rates, NAC provides a way to assess the tumor chemosensitivity and evaluate the mechanisms of resistance to chemotherapy through the evaluation of residual tumor burden. 

Axillary lymph node involvement is the most important prognostic factor in BC, and has long been proven to be correlated with poor survival outcomes [3,4,5,6]. In the neoadjuvant setting, several studies have established the critical role of nodal burden in the assessment of prognosis after NAC in large cohorts of patients [7,8,9,10,11].

Pathologic complete response (pCR) is defined as the absence of invasive cancer in the breast and axillary lymph nodes, and has been shown to be associated with a better long-term survival among BC patients treated with NAC. Although nodal axillary response has been described as a superior prognostic parameter after NAC [12,13], overall pCR is more frequently used and has been adopted by the Food and Drug Administration and the European Medicines Agency as an important endpoint in BC neoadjuvant studies [14]. 

The prognostic value of pCR to predict event-free survival varies among BC subtypes [15,16]. In 2014, a meta-analysis by Cortazar et al. [17] including 11,955 patients found a stronger association between pCR and long-term outcomes in patients with TNBCs (RFS: HR = 0.24, 95% CI [0.18–0.33]) and in those with *HER2*-positive hormone receptor negative BC (RFS: HR = 0.15, 95% CI [0.09–0.27]); whereas the association was less marked in *HER2*-positive hormone receptor positive BC (RFS: HR = 0.58, 95% CI [0.42–0.82]) and luminal BC (RFS: HR = 0.49, 95% CI [0.33–0.71]). 

However, the evidence evaluating the prognostic impact of residual axillary burden after NAC according to BC subtypes is scarce. Most of the studies evaluating the prognostic impact of axillary response to NAC classified patients in a binary manner, depending on the presence or absence of residual nodal disease, without taking into account the number of axillary lymph nodes involved; fewer studies, if any, performed upfront comparison of the prognostic significance by the BC subtype. 

The aim of our study is to evaluate the impact of the number of axillary nodes involved on survival outcomes according to BC subtype in a real-life cohort of breast cancer patients treated with NAC. 

## 2. Results

### 2.1. Baseline Patients’ and Tumors’ Characteristics 

A total of 1197 patients were included in the cohort. Patients’ baseline characteristics are summarized in Table 1. Median age was 48 years old. Patient’s repartition by subtype was as follows: luminal (*n* = 526, 43.9%), TNBC (*n* = 376, 31.4%), *HER2*-positive (*n* = 295; 24.6%). The nodal status of patients at diagnosis was as follows: 525 patients (44%) were node negative before neoadjuvant treatment (*n* = 235 luminal BC (45%); *n* = 171 TNBC (45.5%); *n* = 119 HER-2 positive BC (40.3%)). Out of the 295 HER-2 positive patients, 204 (69.2%) received HER-2 targeted therapy. 

After NAC, 43% of the patients (515/1197) had a nodal involvement. Patients with bigger tumors, with clinical baseline nodal involvement, luminal BCs (versus TNBC or *HER2-* positive), low proliferative tumors (versus high proliferative), with lower immune infiltration (versus high TIL levels) were more likely to have a nodal involvement at NAC completion. Among *HER-2* positive BC patients, those having been treated with trastuzumab were more likely to have no axillary disease after NAC (Appendix A). Axillary staging technique was axillary lymph node dissection for the majority of patients (n = 1169, 97.7%). 

The number of nodes ranged from 1 to 35 (median: 11) (Figure 1A) and the number of lymph nodes involved varied from 0 to 21(Figure 1B). In case of nodal involvement, the median number of nodes involved was 2 (Figure 1C), and the repartition was significantly different among BC subtypes. Overall, 57% of the patients had no nodal involvement at axillar surgery (*n* = 682), 28% had a mild nodal involvement (*n* = 341), and 15% (*n* = 174) had a high nodal involvement (Figure 1D). This repartition was significantly different by BC subtype (*p* < 0.001) (Figure 1E).

### 2.2. Association Between Post-NAC Involvement and Tumor Characteristics 

Among post-NAC characteristics, node positivity was associated with RCB index (Table 2, Appendix A), with the presence of lymphovascular invasion (Appendix A), and with higher post-NAC tumor cellularity (Appendix A). Neither post-NAC mitotic index (Appendix A), stromal (Appendix A) nor IT TILs (Appendix A) were significantly associated with post-NAC nodal status. Similar patterns were observed within each BC subtype (Appendix A), with the very exception of post-NAC tumor cellularity (all three BC subtypes), post-NAC mitotic index (luminal BC), and str TILs levels (*HER2*-positive BC) that were significantly higher with increasing number of nodes involved (Appendix A).

### 2.3. Survival Analyses

With a median follow-up of 110.5 months (118.6 months for luminal BC patients, 102.6 months for TNBC patients, 106.3 months for *HER-2* positive BC patients), 371 patients experienced relapse, and 228 died. After univariate and multivariate analysis, post-NAC nodal involvement was significantly associated with RFS in the whole population (*p* < 0.001) (Table 3). After analyses by BC subtype, the association between nodal involvement binned by 3 classes and RFS was significant in all the BC subgroups, but this association was significantly different according to the BC subtype (*P*_interaction_ = 0.004). In the whole population, mild post-NAC nodal involvement (1 to 3); and high nodal involvement were associated with an impaired RFS after univariate analysis (HR = 1.79, 95% CI [1.42–2.28] and HR = 3.3, 95% CI [2.59–4.32]) and after multivariate analysis (HR = 2.06, 95% CI [1.59–2.66] and HR = 3.6, 95% CI [2.73–4.75]) (Figure 2A). The association between RFS and axillary involvement compared with pCR was also studied (Appendix A). *p* values tended to be lower with nodal status and AIC were systematically lower with nodal involvement (Appendix A).

In luminal BCs, mild post-NAC nodal involvement was not associated with an impaired RFS when compared with ypN0 tumors (HR = 1.24, 95% CI [0.86–1.79] in univariate analysis and HR = 1.18, 95% CI [0.82–1.71] in multivariate analysis) (Appendix A), whereas patients with a high nodal involvement were associated with an adverse prognosis (HR = 2.8, 95% CI [1.93–4.06] in univariate analysis and HR = 2.68, 95% CI [1.84–3.89] in multivariate analysis) (Figure 2B). In TNBCs, both mild (HR = 3.19, 95% CI [2.05–4.98] for univariate analysis, HR = 3.17, 95% CI [2.03–4.95] for multivariate analysis) (Appendix A) and high post-NAC nodal involvement (HR = 4.83, 95% CI [3.06–7.63]; HR = 4.52, 95% CI [2.85–7.17]) were associated with an impaired RFS when compared with ypN0 tumors. The difference between [1,2,3] and 4 or more was statistically significant (*p* < 0.001) (Figure 2C). In *HER2*-positive BCs, patients who had tumors with a mild nodal involvement were at a higher risk of relapse (HR = 2.7, 95% CI [1.64–4.43]) (Appendix A) when compared with node negative tumors, but the prognosis was not significantly different from patients with 4 nodes involved or more (HR = 2.69, 95% CI [1.24–5.8]) (Figure 2D).

There was a significant deviation to the linearity assumption of the association between RFS and post-NAC nodal involvement in the whole population and in the 3 BC subtypes. After statistical modelization, the statistical models best fitted a second-degree polynomial (whole population and luminal subgroup Figure 2E,F respectively), and a restricted cubic spline (TNBC and *HER2*-positive BCs, Figure 2G,H respectively). 

After multivariate analysis (Appendix A), post-NAC nodal involvement was significantly associated with RFS in luminal and TNBCs, but not in *HER2*-positive BC.

Similar results were obtained for the overall survival (Figure 3A–D). The interaction between BC subtype, post-NAC nodal involvement and survival was highly significant (P_interaction_ = 0.005).

## 3. Discussion

In this retrospective study of 1197 BC patients treated with NAC, we confirmed the strong prognostic value of nodal involvement after NAC, and we identified a marked difference in the prognostic impact of the axillar burden among the 3 BC subtypes.

Our study provides several new insights. First, it is in line with the previous reports showing that the prognostic value of the axillary burden outperformed the value of the widely used binary endpoint pathological complete response. Rouzier et al. [12] found a higher correlation between RFS and axillary response to primary chemotherapy than with tumoral breast response in 152 BC patients. Hennessy et al. [13] found no impact of residual breast disease on survival outcomes among patients having achieved axillary pCR in a cohort of 403 BC patients with initial nodal involvement treated with NAC. This was confirmed by Dominici et al. [18] in 2010 in a retrospective study of 102 *HER-2* positive patients. 

Second, along with previous studies (Table 4), we found a higher rate of post-NAC negative nodal status in case of TNBC, *HER2* positive BC, small tumor size, high-grade tumors [10,19,20,21,22,23,24,25], and high Ki67 [26,27]. In 2014, Boughey et al. [20] studied 694 BC patients treated with NAC with clinical nodal involvement at diagnosis. They found significantly higher rates of post-NAC ypN0 status among TNBC and *HER-2* positive BC subgroups (49.4% and 64.7% respectively) than in luminal BC patients (21.1%). In 2016, Mougalian et al. [10] found similar results in a cohort of 1600 stage II/III N+ BC patients: post-NAC negative nodal status rates repartition was 16.4% for luminal BC versus 40.8% for TNBCs and 47.3% for *HER-2* positive BC. 

Our study supports the previous findings that pathological response has different prognostic implications across BC subtypes. It also explores the survival impact of the number of involved nodes after NAC according to BC subtype which has rarely been explored. So far, most studies evaluating the prognostic impact of post-NAC nodal involvement used the binary endpoint ypN0 versus ypN+ [10,12]. Four studies [9,19,20,25] used binned classes approaching the TNM classification (N0; N1: 1 to 3 nodes involved; N2: 4 to 9 nodes involved; N3: 10 or more nodes involved) (Table 4). However, to our knowledge, no study compared upfront the prognostic impact of nodal involvement according to BC subtypes nor performed linearity tests. Our results show that the prognostic value of the number of post-NAC positive nodes differs according to BC subtype. It has been demonstrated that achieving ypT0 and ypN0 in luminal BC is not as predictive of relapse free survival as it is in TNBC or *HER-2* positive BC [17]. Results from our study show that patients with luminal BC presenting post-NAC axillary residual disease up to 3 positive nodes had a similar prognosis to those with no axillary residual disease, while we evidenced a negative impact on survival outcomes when the number of nodes involved was 4 or above. Our data support the argument that a reporting system incorporating this information should be routinely used following NAC. The prognostic impact of low-to-intermediate nodal involvement (1 to 3 nodes involved after chemotherapy) has also been studied in the adjuvant setting. Retrospective analyses from randomized trials have suggested that the recurrence score of a 21 gene assay [28,29] could identify a subset of ER + /HER-2 negative BC patients with positive nodes who did not derive a significant benefit from chemotherapy: Albain et al. [30] and Dowsett et al. [31] found low risks of distant metastases in luminal low recurrence score N+ disease and luminal low recurrence score disease with 1 to 3 nodes involved respectively. The withholding of adjuvant chemotherapy for this category of BC patients is currently being evaluated in an ongoing trial [32]. In the TNBC subgroup, as previously identified by our team [33], a positive nodal status after NAC was a poor prognostic factor, and the prognosis was worsened as soon as one lymph node was involved. However, as shown by the cubic spline statistical model best fitting the data, the slope of the increase of the risk was maximal between 0 and 2 lymph nodes, and the slope decreased thereafter. Finally, in the *HER-2* positive BC subgroup, the existence of residual axillary disease was a poor prognostic factor and the magnitude of the risk was similar for patients with 1 to 3 nodes involved and those with 4 or more nodes involved (RFS HR 2.68 95% CI [1.63–4.41] vs. 2.67 95% CI [1.24–5.77]), though the interpretation might be limited by the weak effective of the latter category (*n* = 20 out of 295 *HER-2* positive BC patients, 6.8%). 

Regarding post-NAC parameters, our study found that node positivity was associated with lymphovascular invasion, which is in line with a previous study describing LVI as a strong prognostic marker [34].

To the best of our knowledge, we report here the first upfront comparison of the prognostic value of residual axillary disease among each BC subtype, while taking into account the number of positive nodes after NAC. In addition, we evidenced that the relationship between nodal involvement and relapse free survival was nonlinear, and this was true in every BC subtype. The main strengths of our study include its large statistical power, its long-term follow-up. Limits of our study include its retrospective design and the absence of external independent validation. It should also be precise that the incidence of missing data was high for several variables (notably LVI, TIL levels, RCB index, Ki 67 and BRCA status), which may have had an impact on the presence or absence of statistically significant associations, although they were removed from multivariate analyses if variables had missing data in more than 30% of the cases. RCB classification data were missing for 40% of the patients, which can be explained by the fact that this study’s cohort predates Fraser Symman’s 2007 paper describing the RCB classification [35]. Pathological response data were extracted from pathology reports and retrospective pathological review of the slides was performed when possible in case of missing data, but was not systematic.

Our study has pragmatic implications. If confirmed in independent studies, it suggests that the cut-off to consider high-risk patients after NAC completion should be different according to BC subtypes: 4 or more nodes involved for luminal BC patients, and 1 for TNBC and *HER-2* positive BC patients. With the widespread routine use of NAC for TNBC and *HER2*-positive BC patients ^1 2^, second-line trials in the post neoadjuvant setting for high risk patients are increasing testing the addition of chemotherapy, PARP inhibitors [36], immunotherapy [37], cyclin-dependent kinase inhibitors [38], or vaccines [39]. We previously demonstrated that TILs-enriched luminal BRCA tumors [40] and TILs-enriched HER-positive BC tumors are at a high risk of relapse [41] and could benefit from additional therapies. In the light of these current scientific developments, residual axillary disease is not a predictive factor of the efficacy of such specific therapies, but our findings are of particular importance since they may help to identify more accurately the high-risk patients who might benefit from such treatments by considering the number of residual positive nodes after NAC as a cornerstone of prognostication, provided that it is interpreted according to histological BC subtype.

## 4. Materials and Methods 

### 4.1. Patients

We analyzed a previously described retrospective cohort of patients [42,43] with invasive breast carcinoma stage T1-T3NxM0 and treated with NAC at Institut Curie, Paris, between 2002 and 2012 (NEOREP Cohort, CNIL declaration number). We included unilateral, non-recurrent, non-inflammatory, non-metastatic tumors, excluding T4 tumors. All patients received NAC, followed by surgery and radiotherapy. NAC regimens changed over our recruitment period (anthracycline-based regimen or sequential anthracycline-taxanes regimen), with trastuzumab used in an adjuvant and/or neoadjuvant setting since 2005. Endocrine therapy (tamoxifen or aromatase inhibitor) was prescribed when indicated. The study was approved by the Breast Cancer Study Group of Institut Curie and was conducted according to institutional and ethical rules concerning research on tissue specimens and patients. Informed consent from patients was not required by French regulations.

### 4.2. Tumor Samples and Pathological Review

#### 4.2.1. BC Subtypes

Cases were considered estrogen receptor (ER) or progesterone receptor (PR) positive (+) if at least 10% of the tumor cells expressed estrogen and/or progesterone receptors (ER/PR), in accordance with the guidelines used in France [44]. HER2 expression was determined by immunohistochemistry with scoring in accordance with the American Society of Clinical Oncology (ASCO)/College of American Pathologists (CAP) guidelines [45]. Scores 3+ were reported as positive, score 1+/0 as negative (-). Tumors with scores 2+ were further tested by FISH. *HER2* gene amplification was defined in accordance with ASCO/CAP guidelines. We evaluated a mean of 40 tumor cells per sample and the mean HER2 signals per nuclei was calculated: a HER2/CEN17 ratio ≥ 2 was considered positive, and a ratio < 2 negative. BC subtypes were defined as follows: tumors positive for either ER or PR, and negative for HER2 were classified as luminal; tumors positive for *HER2* were considered to be *HER2*-positive BC; tumors negative for ER, PR, and *HER2* were considered to be triple-negative breast cancers (TNBC). Tumor cellularity was defined as the percentage of tumor cells (in situ and invasive) on the specimen (biopsy or surgical specimen). Mitotic index was reported per 10 high power fields (HPF) (1 HPF = 0.301 mm^2^).

#### 4.2.2. Post-NAC Nodal Involvement (ypN)

Post-NAC nodal involvement (ypN) was divided into three categories: no axillary involvement (ypN = 0), intermediate involvement (1 to 3 nodes involved, 1 ≤ ypN ≤ 3), and high axillary involvement (4 or more nodes involved, ypN ≥ 4). Nodal extent was also analyzed as a continuous variable. 

#### 4.2.3. Residual Cancer Burden Index (RCB)

Histological components of the “Residual Cancer Burden” were retrieved for calculating the score as described in 2007 by Symmans [35]. RCB index enables the classification of residual disease into four categories: RCB-0 (complete pathologic response = pCR), RCB-I (minimal residual disease), RCB-II (moderate residual disease), and RCB-III (extensive residual disease). RCB was calculated through the web-based calculator that is freely available on the Internet (www.mdanderson.org/breastcancer_RCB).

#### 4.2.4. TILs and LVI

Lymphovascular invasion (LVI) was defined as the presence of carcinoma cells within a finite endothelial-lined space (a lymphatic or blood vessel). Tumor infiltrating lymphocytes (TILs) were defined as the presence of mononuclear cells infiltrate (including lymphocytes and plasma cells, excluding polymorphonuclear leukocytes), and were also evaluated retrospectively for research purposes, according to the recommendations of the international TILs Working Group [46,47].

### 4.3. Study Endpoints 

Relapse-free survival (RFS) was defined as the time from surgery to death, loco-regional recurrence or distant recurrence, whichever occurred first, and overall survival (OS) was defined as the time from surgery to death. Patients for whom none of these events were recorded were censored at the date of their last known contact. Survival cutoff date analysis was 1 February 2019.

### 4.4. Statistical Analysis

The study population was described in terms of frequencies for qualitative variables, or medians and associated ranges for quantitative variables. Chi-square tests were performed to search for differences between subgroups for each variable (considered significant for *p*-values ≤ 0.05). Survival probabilities were estimated by the Kaplan–Meier method, and survival curves were compared in log-rank tests. Hazard ratios and their 95% confidence intervals were calculated with the Cox proportional hazards model. Variables with a *p*-value for the likelihood ratio test equal to 0.05 or lower in univariate analysis were selected for inclusion in the multivariate analysis. A forward stepwise selection procedure was used to establish the final multivariate model and the significance threshold was 5%. For variables that were significantly correlated, collinearity was avoided by retaining only one variable, based on its clinical relevance or likelihood ratio. Variables with missing data in more than 30% of the cases were removed from the multivariate analyses.

### 4.5. Linearity and Interaction Tests

We investigated the linearity of the association between nodal involvement and RFS/OS by comparing the model in which nodal involvement was considered to vary linearly with models based on restricted cubic spline fits and fractional polynomials, as previously described [48]. If significant deviation from the assumption of linearity was observed, based on the lowest AIC of the model, the variable was modeled with ypN binned into classes.

We tested the hypothesis of potentially different effects of ypN in different BC subtypes, by including interaction terms in the Cox model. A *p*-value of 0.10 was selected to determine the statistical significance of the interaction term, as it has been suggested because of a low power of the test in the interaction setting [49]. 

Data were processed and statistical analyses were carried out with R software version 3.1.2 (www.cran.r-project.org, R Foundation for Statistical Computing, 2009). 

## 5. Conclusions

The prognostic value of residual axillary burden differs according to BC subtype. The number of residual positive nodes after NAC should be interpreted according to histological BC subtype to accurately stratify patients with a high risk of recurrence after NAC who should be offered second line therapies.

## Figures and Tables

**Figure 1 cancers-13-00171-f001:**
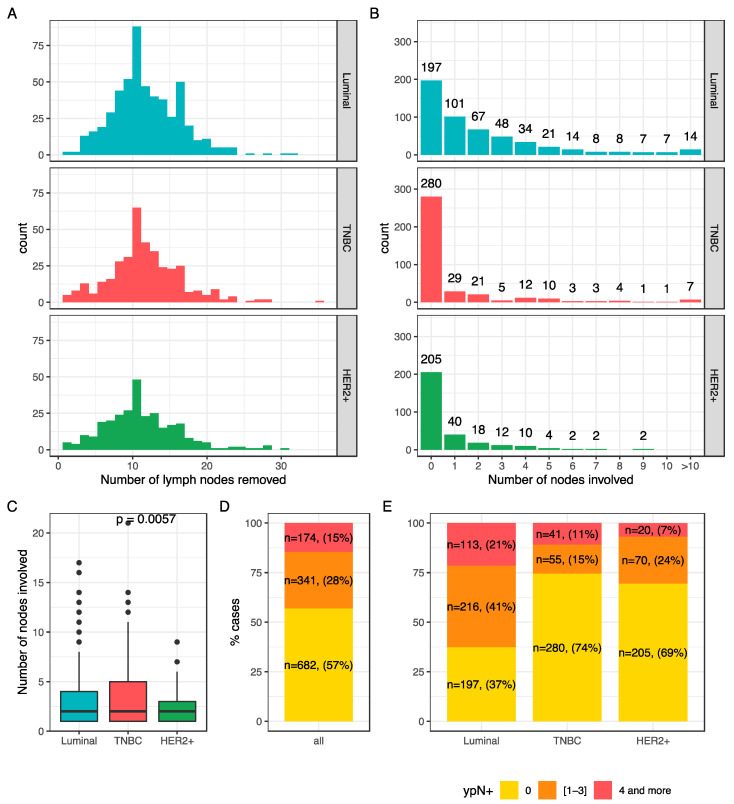
Nodal burden after NAC: number of lymph nodes removed according to BC subtype (**A**); number of involved nodes according to BC subtype (**B**); mean number of nodes involved after NAC according to BC subtype (**C**); node involvement repartition after NAC in the whole population (**D**) and according to BC subtype (**E**).

**Figure 2 cancers-13-00171-f002:**
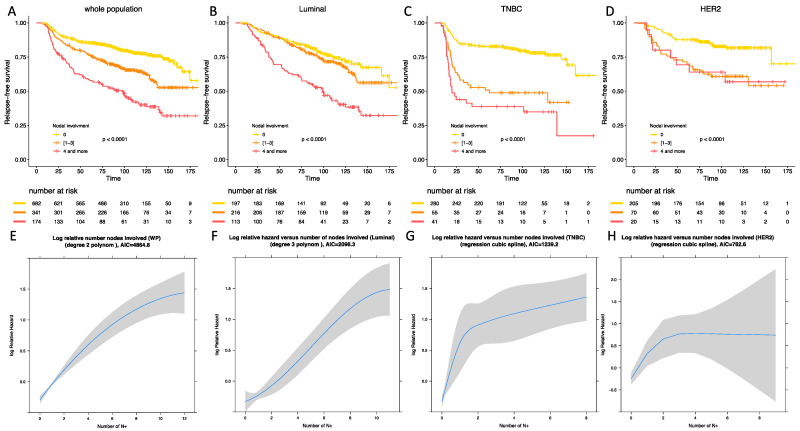
Relapse-free survival according to BC subtype in the whole population (**A**), in luminal BC (**B**), in TNBC (**C**), in *HER-2* positive BC (**D**). Statistical models reflecting the association between relapse free survival and nodal status in the whole population (**E**), in luminal BC (**F**), in TNBC (**G**), and in *HER-2* positive BC (**H**).

**Figure 3 cancers-13-00171-f003:**
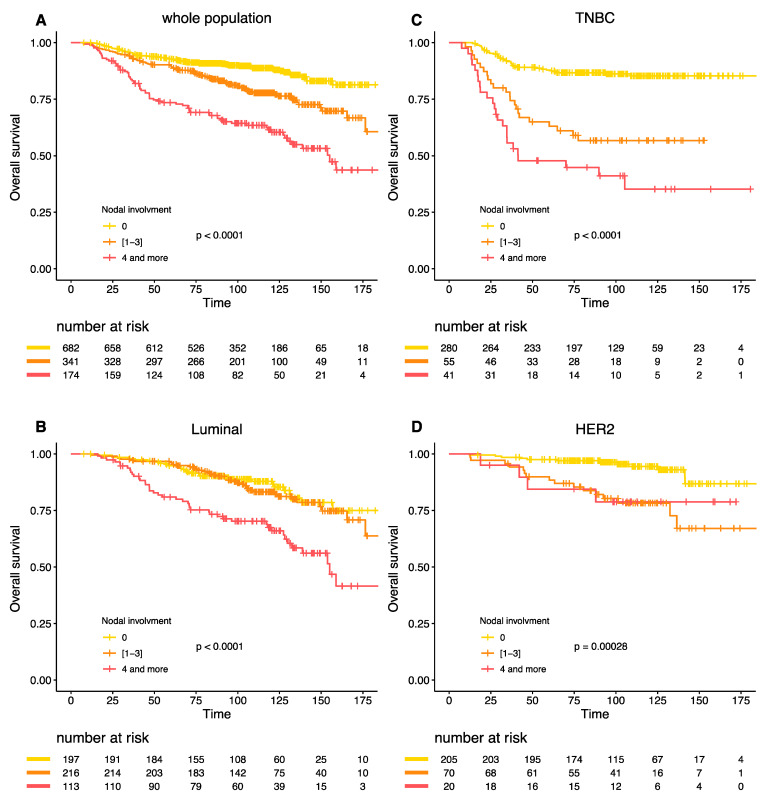
Overall survival according to post NAC nodal involvement in the whole population (**A**), Luminal BC (**B**), TNBC (**C**), and *HER-2* positive BC (**D**).

**Table 1 cancers-13-00171-t001:** Patients and tumor characteristics by post- neoadjuvant chemotherapy (NAC) nodal involvement.

Characteristics	Class	All	Node Negative	Node Positive	*p*
*n*		1197 (100%)	682 (57%)	515 (43%)	
Age median		48.5 (10.1)	48.0 (10.4)	49.3 (9.6)	0.027
Age class	[0–50)	678 (56.6)	399 (58.5)	279 (54.2)	0.279
	[50–60)	352 (29.4)	189 (27.7)	163 (31.7)	
	60+	167 (14.0)	94 (13.8)	73 (14.2)	
Menopausal status	Premenopausal	746 (62.8)	432 (63.9)	314 (61.3)	0.396
	Postmenopausal	442 (37.2)	244 (36.1)	198 (38.7)	
BMI	18.5–24.9	680 (57.1)	401 (59.1)	279 (54.5)	0.302
	<18.5	48 (4.0)	26 (3.8)	22 (4.3)	
	25–29.9	304 (25.5)	160 (23.6)	144 (28.1)	
	>=30	159 (13.4)	92 (13.5)	67 (13.1)	
Smoking status	No	719 (75.6)	428 (76.4)	291 (74.4)	0.528
	Yes	232 (24.4)	132 (23.6)	100 (25.6)	
BRCA mutation genes	BRCA1	31 (11.7)	24 (13.7)	7 (7.8)	0.404
	BRCA2	14 (5.3)	10 (5.7)	4 (4.4)	
	others	1 (0.4)	1 (0.6)	0 (0.0)	
	No	219 (82.6)	140 (80.0)	79 (87.8)	
Clinical T stage (TNM)	T0-T1	70 (5.9)	41 (6.0)	29 (5.6)	0.001
	T2	797 (66.6)	483 (70.8)	314 (61.1)	
	T3-T4	329 (27.5)	158 (23.2)	171 (33.3)	
Clinical N stage (TNM)	N0	525 (43.9)	372 (54.5)	153 (29.8)	<0.001
	N1-N2-N3	671 (56.1)	310 (45.5)	361 (70.2)	
BC subtype	Luminal	526 (43.9)	197 (28.9)	329 (63.9)	<0.001
	TNBC	376 (31.4)	280 (41.1)	96 (18.6)	
	HER2+	295 (24.6)	205 (30.1)	90 (17.5)	
ER status	Negative	544 (45.4)	399 (58.5)	145 (28.2)	<0.001
	Positive	653 (54.6)	283 (41.5)	370 (71.8)	
PR status	Negative	680 (57.9)	450 (66.8)	230 (46.0)	<0.001
	Positive	494 (42.1)	224 (33.2)	270 (54.0)	
Her2 status	Negative	902 (75.4)	477 (69.9)	425 (82.5)	<0.001
	Positive	295 (24.6)	205 (30.1)	90 (17.5)	
Histological type	NST	1060 (93.5)	617 (96.0)	443 (90.2)	<0.001
	Others	74 (6.5)	26 (4.0)	48 (9.8)	
KI67	[0–10)	65 (11.2)	25 (7.9)	40 (15.0)	0.001
	[10–20)	110 (18.9)	49 (15.6)	61 (22.9)	
	≥20	406 (69.9)	241 (76.5)	165 (62.0)	
Mitotic index	≤22	20.8 (19.2)	24.1 (20.2)	16.3 (16.9)	<0.001
	>22	61.3 (18.2)	61.0 (19.2)	61.8 (16.4)	
SBR grade	Grade I-II	477 (41.3)	200 (30.4)	277 (55.7)	<0.001
	Grade III	678 (58.7)	458 (69.6)	220 (44.3)	
LVI	No	267 (61.0)	158 (69.9)	109 (51.4)	<0.001
	Yes	171 (39.0)	68 (30.1)	103 (48.6)	
DCIS component	No	601 (60.7)	385 (66.3)	216 (52.8)	<0.001
	Yes	389 (39.3)	196 (33.7)	193 (47.2)	
Stromal TIL levels (%)		24.0 (19.9)	26.9 (21.7)	19.3 (15.6)	<0.001
Intra Tumoral TIL levels (%)		11.2 (12.3)	12.4 (13.1)	9.3 (10.6)	0.001
CT regimen (NAC)	anthra-taxanes	841 (70.6)	507 (74.7)	334 (65.1)	<0.001
	anthra	235 (19.7)	105 (15.5)	130 (25.3)	
	taxanes	25 (2.1)	13 (1.9)	12 (2.3)	
	others	91 (7.6)	54 (8.0)	37 (7.2)	
Adjuvant chemotherapy	No	1000 (83.5)	661 (96.9)	339 (65.8)	<0.001
	Yes	197 (16.5)	21 (3.1)	176 (34.2)	
	5FU and Vinorelbine	144 (73.1)	8 (38.1)	136 (77.3)	<0.001
	Others	53 (26.9)	13 (61.9)	40 (22.7)	
Adjuvant Anti HER2 therapy	No	166 (39.2)	18 (9)	148 (66.1)	<0.001
	Yes	258 (60.8)	182 (91)	76 (33.9)	
Radiotherapy	No	22 (1.8)	15 (2.2)	7 (1.4)	0.393
	Yes	1175 (98.2)	667 (97.8)	508 (98.6)	
Infraclavicular Radiotherapy	No	203 (17.3)	161 (24.3)	42 (8.3)	<0.001
	Yes	969 (82.7)	503 (75.7)	466 (91.7)	
Sus-clavicular Radiotherapy	No	121 (10.3)	113 (17)	8 (1.6)	<0.001
	Yes	1051 (89.7)	551 (83)	500 (98.4)	
Axillar Radiotherapy	No	947 (80.8)	623 (93.8)	324 (63.8)	<0.001
	Yes	225 (19.2)	41 (6.2)	184 (36.2)	
Endocrine therapy	No	544 (45.4)	404 (59.2)	140 (27.2)	<0.001
	Yes	653 (54.6)	278 (40.8)	375 (72.8)	
Type of endocrine therapy	Tamoxifen	397 (33.3)	178 (26.3)	219 (42.5)	<0.001
	Aromatase inhibitor	217 (18.2)	82 (12.1)	135 (26.2)	
	Tamoxifen + Agonist	12 (1)	5 (0.7)	7 (1.4)	
	Aromatase inh + Ag	6 (0.5)	3 (0.4)	3 (0.6)	
	Others	20 (1.7)	9 (1.3)	11 (2.1)	

Abbreviations: BMI = body mass index; NST = no special type; ER = estrogen receptor; PR = progesterone receptor; NAC = neoadjuvant chemotherapy; CT = chemotherapy; AC = anthracyclines; TILs = tumor-infiltrating lymphocytes; RCB = residual cancer burden; LVI = lymphovascular invasion. Missing data: menopausal status, *n* = 9; BMI, *n* = 6; smoking status, *n* = 246; BRCA mutation genes, *n* = 932; clinical T stage (TNM), *n* = 1; clinical N stage (TNM), *n* = 1; PR status, *n* = 23; histological type, *n* = 63; KI67, *n* = 616; mitotic index, *n* = 117; NA, *n* = 484; SBR grade, *n* = 42; LVI, *n* = 759; DCIS component, *n* = 207; stromal TIL levels (%), *n* = 483; IT TIL levels (%), *n* = 483; CT regimen (NAC), *n* = 5^1^ The “*n*” denotes the number of patients. In case of categorical variables, percentages are expressed between brackets. In case of continuous variables, mean value is reported, with standard deviation between brackets. In case of non-normal continuous variables, median value is reported, with interquartile range between brackets.

**Table 2 cancers-13-00171-t002:** Tumor characteristics by post-NAC nodal involvement.

Characteristics	Class	Post-NAC Node Involvement (ypN)			*p*
		0	1–3	4 and More	
*n*		682	341	174	
Pathological complete response	No pCR	396 (58.3)	340 (100.0)	173 (100.0)	<0.001
	pCR	283 (41.7)	0 (0.0)	0 (0.0)	
Post-NAC LVI	No	318 (78.3)	156 (55.5)	56 (38.6)	<0.001
	Yes	88 (21.7)	125 (44.5)	89 (61.4)	
RCB index (continuous)		1.0 (0.9)	3.0 (0.8)	3.6 (0.7)	<0.001
RCB class	RCB-0	200 (45.1)	0 (0.0)	0 (0.0)	<0.001
	RCB-I	53 (12.0)	12 (6.4)	0 (0.0)	
	RCB-II	188 (42.4)	101 (53.7)	20 (24.1)	
	RCB-III	2 (0.5)	75 (39.9)	63 (75.9)	
Stromal TIL levels (%) (post-NAC)		12.8 (13.1)	13.6 (12.3)	12.6 (12.1)	0.750
IT TIL levels (%) (post-NAC)		7.2 (8.2)	6.9 (8.0)	5.7 (5.4)	0.289
Mitotic index (post-NAC)		18.9 (30.9)	12.6 (23.0)	16.9 (34.8)	0.103
Tumor cellularity (post-NAC)		19.6 (26.7)	35.9 (25.5)	36.5 (24.5)	<0.001

Missing data: pathological complete response, *n* = 5; post-NAC LVI, *n* = 365; RCB index (continuous), *n* = 483; RCB class: 0; RCB-0; [0;1.36]: RCB-I; [1.36–3.28]: RCB-II; >=3.28: RCB-III, *n* = 483; stromal TIL levels (%) (post-NAC), *n* = 483; IT TIL levels (%) (post-NAC), *n* = 715; mitotic index (post-NAC), *n* = 722; tumor cellularity (post-NAC), *n* = 483. The “*n*” denotes the number of patients. In case of categorical variables, percentages are expressed between brackets. In case of continuous variables, mean value is reported, with standard deviation between brackets. In case of nonnormal continuous variables, median value is reported, with interquartile range between brackets.

**Table 3 cancers-13-00171-t003:** Association of clinical and pathological pre and post-NAC parameters with relapse-free survival after univariate and multivariate analysis in the whole population.

				Univariate				Multivariate		
Variable	Category	*n*	Events	HR	95% CI	*p **	*p*	HR	95% CI	*p*
Pre-NAC parameters										
Age	[0–50)	678	210	1			0.716			
	[50–60)	352	106	0.97	[0.77–1.22]					
	60+	167	55	1.11	[0.82–1.49]					
Menopausal status	Pre	746	232	1			0.87			
	Post	442	135	0.98	[0.79–1.21]					
BMI	18.5–24.9	680	193	1			0.009	1		
	<18.5	48	15	1.13	[0.67–1.91]	0.651		1.13	[0.66–1.91]	0.66
	25–29.9	304	96	1.15	[0.9–1.47]	0.255		1.06	[0.83–1.36]	0.624
	>= 30	159	66	1.63	[1.23–2.15]	<0.001		1.52	[1.15–2.02]	0.003
Smoking status	No	721	221	1			0.924			
	Yes	233	70	0.99	[0.75–1.29]					
BRCA mutation genes	BRCA1	31	9	1			0.991			
	BRCA2	14	4	0.91	[0.28–2.96]					
	others	1	0							
	No	220	59	0.89	[0.44–1.79]					
Clinical T stage (TNM)	T0-T1	70	18	1			<0.001	1		
	T2	797	223	1.1	[0.68–1.78]	0.703		1.25	[0.77–2.03]	0.371
	T3-T4	329	129	1.79	[1.09–2.93]	0.021		1.69	[1.02–2.78]	0.04
Clinical N stage (TNM)	N0	525	148	1			0.032			
	N1-N2-N3	671	223	1.26	[1.02–1.55]					
BC subtype	Luminal	526	184	1			0.025	1		
	TNBC	376	116	1.05	[0.83–1.33]	0.668		1.66	[1.29–2.15]	<0.001
	HER2 +	295	71	0.72	[0.54–0.94]	0.017		1.04	[0.78–1.39]	0.785
ER status	Negative	544	158	1			0.953			
	Positive	653	213	0.99	[0.81–1.22]					
PR status	Negative	680	208	1			0.288			
	Positive	494	152	0.89	[0.72–1.1]					
Her2 status	Negative	902	300	1			0.007			
	Positive	295	71	0.7	[0.54–0.91]					
Histological type	NST	1060	317	1			0.106			
	Others	74	30	1.36	[0.94–1.98]					
KI67	[0–10)	65	21	1			0.494			
	[10–20)	110	38	1.07	[0.63–1.82]					
	≥20	406	144	1.25	[0.79–1.98]					
SBR grade	Grade I-II	477	170	1			0.11			
	Grade III	678	188	0.84	[0.69–1.04]					
LVI	No	267	98	1			0.63			
	Yes	171	66	1.08	[0.79–1.48]					
DCIS component	No	604	165	1			0.11			
	Yes	389	135	1.2	[0.96–1.51]					
CT regimen (NAC)	anthra-taxanes	845	234	1			0.017			
	anthra	235	97	1.37	[1.07–1.74]	0.011				
	taxanes	25	4	0.59	[0.22–1.59]	0.3				
	others	91	36	1.42	[1–2.02]	0.052				
Post-NAC parameters										
pCR	No pCR	911	332	1			<0.001			
	pCR	285	39	0.35	[0.25–0.49]	<0.001				
Post-NAC LVI	No	531	143	1			<0.001			
	Yes	302	144	2	[1.59–2.52]	<0.001				
ypN	0	682	144	1			<0.001	1	-	-
	[1,2,3]	341	127	1.8	[1.42–2.28]	<0.001		2.06	[1.59–2.66]	<0.001
	4 and more	174	100	3.35	[2.59–4.32]	<0.001		3.6	[2.73–4.75]	<0.001
RCB class	RCB-0	202	23	1			<0.001			
	RCB-I	65	7	0.98	[0.42–2.29]	0.965				
	RCB-II	309	102	3.24	[2.06–5.09]	<0.001				
	RCB-III	141	72	5.56	[3.47–8.89]	<0.001				

Abbreviations: pCR = pathological complete response; BMI = body mass index; NST = no special type; ER = estrogen receptor; PR = progesterone receptor; NAC = neoadjuvant chemotherapy; AC = anthracyclines; TILs = tumor infiltrating lymphocytes; RCB = residual cancer burden; LVI = lymphovascular invasion. *p* represents the *p*-value for the Wald test, and *p* * represents the individual *p*-value versus reference class.

**Table 4 cancers-13-00171-t004:** Summary of previous studies comparing prognosis according to nodal involvement after neoadjuvant chemotherapy (NAC) according to breast cancer subtype.

Study	Study Population	Study Design	Number of Patients				Median f-u (mo.)	Post-C Nodal Involvement WP (*n*, %)	Post-NAC Nodal Involvement According to BC Subtype (*n*, %)			5 Years RFS WP (HR)	Interaction Test ypN/BC Subtype
			*n*	HR+/HER2- (%)	TNBC (%)	HER2+ (%)			HR+/HER2- (%)	TNBC (%)	HER2+ (%)		
**McReady** (1989) *Archives of Surgery*	T3-T4, N2-N3 BC	RA	136	-	-	-	56	None (*n* = 34, 25%) 1–3 (*n* = 43, 32%) 4–10 (*n* = 35, 26%) >10 (*n* = 24, 17%)	-	-	-	-	-
**Kuerer** (1998) *The American Journal of Surgery*	IIA, IIB, IIIA, IIIB, IV BC	CT	165	-	-	-	35	None (*n* = 49, 30%) 1–3 (*n* = 51, 31%) 4–10 (*n* = 43, 27%) >10 (*n* = 20, 12%)	-	-	-	-	-
**Kuerer** (1999) *Annals of Surgery*	N+ IIA, IIB, IIIA, IIIB, IV BC	CT	191	-	-	-	61	None (*n* = 43, 23%) ≥1 (*n* = 148, 77%)	-	-	-	-	-
**Pierga** (2000) *British Journal of Cancer*	T2–T3, N0–N1 BC	RA	487	-	-	-	84	None (*n* = 223, 45.8%) 1–3 (*n* = 159, 32.6%) 4–7 (*n* = 72, 14.8%) ≥8 (*n* = 34, 7%)	-	-	-	1 1.6 [1.2–2.3] 2.3 [1.5–3.4] 6.3 [4.1–9.7]	-
**Rouzier** (2002) *JCO*	T1-T3 N+ BC	RA	152	-	-	-	75	ypN0 (*n* = 35, 23%) ypN+ (*n* = 117, 77%)	-	-	-	1 3.4 [2–5.9]	-
**Hennessy** (2005) *JCO*	stage II/III N+ BC	CT	403	-	-	-	64	ypN0 (*n* = 89, 22%) ypN+ (*n* = 314, 78%)	-	-	-	-	-
**Dominici** (2010) *Cancer*	T1-T4 N+ BC	RA	109	0	0	109	29	ypN0 (*n* = 81, 74%) ypN+ (*n* = 28, 26%)	-	-	74% 26%	-	-
**Zhang** (2013) *Curr. Oncol.*	stage II/III	RA	301	145 (48.2%)	55 (18.3%)	101 (33.6%)	36.2	ypN0 pCR (*n* = 75, 24.9%) ypN0 non pCR (*n* = 103, 34.2%) ypN1 (*n* = 72, 23.9%) ypN2 (*n* = 35, 11.6%) ypN3 (*n* = 16, 5.3%)	11.7% 34.5% 31.7% 17.2% 4.8%	25.4% 43.6% 18.2% 5.5% 7.3%	43.5% 28.7% 15.8% 6.9% 5%	0.07 0.53 1 5.51 3.8	-
**Boughey** (2014) *Ann Surg*	T1-T4 N1-2 M0 BC	CT	694	317 45.7%)	170 (24.5%)	207 29.8%)	-	ypN0 (*n* = 285, 41.1%) ypN1 (*n* = 241, 34.7%) ypN2 (*n* = 129, 18.6%) ypN3 (*n* = 39, 5.6%)	21.1% 43% 27.4% 9.5%	49.4% 32.4% 15.3% 2.9%	64.7% 25.6% 7.7% 1.9%	-	-
**Kim** (2015) *Medicine (Baltimore)*	T1-T4 N1-3 M0 BC	RA	415	245 (59%)	93 (22.4%)	77 (18.6%)	-	ypN0 (*n* = 159, 38.3%) ypN+ (*n* = 256, 61.7%)	29% 71%	53.8% 46.2%	49.4% 50.6%	-	-
**Bonsang-Kitzis** (2015) *PLoS One*	T1-T3 N1-3 M0 BC	RA	326	-	326 (100%)	-	52	ypN0 (*n* = 245, 75%) ypN+ (*n* = 81, 25%)	-	75% 25%	-	1 3.48 [2.08–5.84]	-
**Mougalian** (2016) *JAMA Oncology*	Stage II/III N+ BC	RA	1600	719 (53.42%)	289 (21.47%)	338 (25.1%)	79	ypN0 (*n* = 454, 28.4%) ypN+ (*n* = 1146, 71.6%)	16.4% 83.6%	40.8% 59.2%	47.3% 52.7%	1 3.1 [2.3–4.15] *	-
**Mamtani** (2016) *Ann Surg Oncol*	Stage II/III N+ BC	CT	195	73 (37.4%)	55 (28.2%)	67 (34.4%)	-	ypN0 (*n* = 96, 49%)	21%	47%	82%	-	-
**Al-Tweigeri** (2016) *Cancer Chemot Parmacol*	T2–T4, N0–N2 M0 BC	CT	80	38 (47.5%)	13 (16.5%)	29 (36%)	43	ypN0 (*n* = 51, 63.7%)	50%	73%	79%	-	-
**Diego** (2016) *Ann Surg Oncol*	Stage II/III N+ BC	RA	30	2 (7%)	12 (36%)	16 (57%)	-	ypN0 (*n* = 19, 63%)	0%	67%	69%	-	-
**Boland** (2017) *BJS Open*	T1-T4 N+ BC	RA	284	154 (54.2%)	30 (10%)	102 (35.9%)		0 (*n* = 105, 37%) 1 (*n* = 41, 14.4%) 2–4 (*n* = 63, 22.2%) 5–10 (*n* = 43, 15.1%) >10 (*n* = 29, 10.2%)	22.7% 14.9% 26.6% 20.8% 14.3%	50% 6.6% 16.7% 10% 13.3%	54% 15.7% 16.7% 7.8% 2.9%	-	-
**Our study** (2020)	T1-T3 NxM0 BC	RA	1197	526 (43.9%)	376 (31.4%)	295 (24.6%)	110.5	0 (*n* = 682, 57%) 1–3 (*n* = 341, 28%) ≥4 (*n* = 174, 15%)	37% 41% 21%	74% 15% 11%	69% 24% 7%	1 1.79 [1.41–2.28] 3.3 [2.56–4.27] **	P_interaction_ = 0.004

Abbreviations: RA = retrospective analysis; CT = clinical trial; RFS = Relapse free survival; * 5 years RFS in the *HER2* BC subgroup: ypN0 = 1; ypN+ = 4.51 [2.7–7.4]; ** 5 years RFS in the luminal BC subgroup: 0 = 1; 1–3 = 1.24 [0.86–1.79]; ≥4 = 2.8 [1.93–4.06] / 5 years RFS in the TNBC subgroup: 0 = 1; 1–3 = 3.23 [2.07–5.05]; ≥4 = 4.67 [2.94–7.42]/5 years RFS in the *HER2* BC subgroup: 0 = 1; 1–3 = 2.68 [1.63–4.41]; ≥4 = 2.67 [1.24–5.77].

## Data Availability

The data presented in this study are available in the present manuscript or in the Appendix A section.

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
