# Peer review of "The Prognostic Value of Lymph Node Involvement after Neoadjuvant Chemotherapy Is Different among Breast Cancer Subtypes"

_cancers, 2021, doi:10.3390/cancers13020171_

Round 1

Reviewer 1 Report

Laot et al. described an analysis from a prospective cohort on the prognostic value of lymph node numbers / N category after neoadjuvant chemotherapy. In this manuscript the authors claimed that threshold of involved lymph nodes after NAC to predict shorter relapse free survival is different among three breast cancer subtypes, with a higher threshold (N2 / >=4 positive nodes) for luminal subtypes. While this is an interesting work from a strong database, there are issues to be further clarified before the work can be evaluated.

One of the concerns is the analysis design. It is difficult to interpret RFS analysis in a real world cohort without information on surgery, radiotherapy, hormone therapy or targeted therapy. Especially it is not clear how many patients with Her2+ disease receive anti-Her2 therapy in the neoadjuvant setting. If only <10% of patients received anti-Her2 in their neoadjuvant chemo (NAC) in this cohort, than it should be clearly pointed out to avoid misinterpretation in the modern era. This may directly impact on the threshold for ypN to predict a shorter RFS in Her2+ disease according to the standard practice today.

The second concern is about the statistical tool used. The authors put a lot of emphasis to interpret soft correlation from univariate analysis, but relatively less effort is paid to discuss results from multivariate analysis. The statistical method to analyze interaction between ypN category and BCa subtype was not found in the relevant methods section either.

Moreover, does the multivariate analysis consider the residual tumor size as a variable, and how does it interact with 3-tier ypN system to predict RFS among different disease subtypes?

The third concern. In the text it was claimed RFS interacts with ypN in luminal disease and TNBC but not Her2-enriched, the data it referred to in the supplement tables was on the “overall-free survival” but not RFS. Even it was a mislabeling, the ypN threshold for Her2-enriched disease is uncertain. Therefore it is not based on the authors’ data to claim that 3-tier ypN predicts RFS differently between luminal and Her2-enriched diseases as presented in the abstract and in the most of the main text.

There are additional minor comments.
1. The BRCA results behave differently to the main population. How are these patients selected for BRCA analysis, and how do they respond to treatment?
2. In the discussion Line 167: it is not well established that axillary burden outperforms the binary endpoint of pCR. The author may want to add an analysis to test this claim with their data.

  1. The table 4 is very difficult to read. Also the authors’ claim that their study is the only one that has interaction test with breast cancer subtype is not solid.
  2. How many patients receive add-on chemotherapy after definitive treatment? Does it interfere with RFS interpretation?
  3. Conclusion Line 223-228: details of trials can be cited in the reference but not listed here.
  4. The reference list needs to be polished. For example, “J Clin Oncol Off J Am Soc Clin Oncol” apparantly is not an adequate format for JCO. Also please check with the editorial office to see if reference to web-based material is presented in the adequate format.

Author Response

We thank reviewer #1 for the important comments and the essential revisions he/she suggested. Please find enclosed our response to reviewer 1. 

Reviewer 2 Report

The paper by Laot et al is a large retrospective study showing the prognostic value of post NAC residual axillary disease on the prognosis of breast cancer patients. The authors have quantified the residual axillary disease, the novelty of the study is also that the authors stratified the patients according to different breast cancer subtypes. The research is well conducted, includes a large number of patients and the data analysis is comprehensive and sound. I believe that the study as such is a welcome contribution to Cancers. I only have a few minor comments as specified below.

  • Table 1:It is not clear how the mitotic index is categorised. Please also explain all the abbreviations used (TIL, CT, NST)
  • It should be added to the discussion that despite retrospective design, the incidence of missing data was high for many variables (although not for primary outcome) and this should be acknowledged. There were several variables there were considered statistically significant (post-NAC LVI, RCB, etc.) and the number of missing cases was very high. This should be acknowledged. The same is valid for survival analysis and Table 3. There are a lot of missing cases (e.g. BRCA status) and it could be that the significance was absent due to the missing cases.

Author Response

We would like to thank reviewer #2 for his/her interesting comments. Please find enclosed our response to reviewer #2. 

Reviewer 3 Report

This is a large retrospective study on the prognostic significance of post-NAC axillary nodal involvement for breast cancer patients differentiated according to subtypes and number of involved lymph-nodes.

The subject is of great clinical interest and can have impact on pragmatic clinical decisions (e.g. post-neoadjvuant treatment).

I congratulate the authors to this very large and detailed analysis und the abundant data analyzed and presented as well as to the extensive statistics.

I strongly suggest that this paper should be considered for publication, in general.

However, before publication is possible, several aspects have to addressed by the authors:

minor revisions:

p.1 l.44: it is stated that NAC is estalished after publication of the CREATE-X and KATHERINE-trial. Of course, this is not true or at least misleading. These trials established post-neoadjuvant treatment options. There are numerous clinical trials citable between 2002 and 2012 demonstrating benefit of NAC for TNBC and HER2+ disease.

Major revisions:

in general, the presented data is so extensive that the authors should also consider to divide the work into two manuscripts.

the authors must provide a differentiation whether  radiotherapy of the axillary/ infraclavicular/ supraclavicular area was performed. Is there a difference in the prognostic analyses? This is clinically relevant and must be very clear in order to draw further conclusions.

the authors should provide further information on the kind and number of adjuvant treatment regimens.

the authors should  discuss the prognostic impact of LVI post-NAC in their study.

In the light of current scientific developments and clinical decisions (post-neoadvuvant options) :

The authors should include a discussion of the following clinical/ histopathological parameters: BRCAmut, TIL levels.

I would very much appreciate to see the revised version of this very impressive work.

Author Response

We would like to thank reviewer #3 for the important comments and the essential revisions he/she suggested. Please find enclosed in the attached document our response to reviewer #3. 

Reviewer 4 Report

Thank you for asking me to review this interesting manuscript, which looks at the prognostic value of residual axillary lymph node disease following neoadjuvant chemotherapy, according to breast cancer subtype.

This is a well-written manuscript which provides further data to support the hypothesis that pathological complete response following neoadjuvant chemotherapy (ypT0 ypN0) is not equally prognostic across all subtypes. The findings are not surprising, given that the Cortazar meta-analysis cited by the authors demonstrated that ypT0 ypN0 response was better prognostically than ypT0 alone, and that the association between pCR and outcome was strongest in TNBC and HER2 positive disease. It is strengthened by a relatively large cohort of patients with a long follow up, albeit from a single centre.

I have a few comments/questions for the authors:

I’m aware that the authors have previously reported this patient cohort, but they need to provide some brief details of the cohort in this publication. These are as follows:

  • Were all the patients in this cohort node positive at diagnosis? How was the axilla staged at baseline (i.e. pre-neoadjuvant systemic therapy)? Are there patients included in the cohort who were node negative prior to commencing treatment? If so, by definition, they could not have had residual disease in the axilla post-chemotherapy.
  • Did all patients have axillary node dissection to establish their post-chemotherapy nodal status? Or did some of them have SLNB? If having SLNB then the proportion with positive and negative SLNB and those having completion ALND should be provided.
  • Was pathological response data obtained from review of pathology reports or was there pathologist review of the slides?
  • Please state median follow up, both for the cohort as a whole and in each subtype (if there is a difference). Recurrence is much more likely to occur as a late event in the luminal group.

RCB classification data is missing for 40% of cases – presumably because this patient cohort predates Fraser Symman’s 2007 paper describing the RCB. However, this should be explicitly acknowledged in the text.

In their discussion section, the authors should consider the fact that the RCB system incorporates nodal response information. What, therefore, do they consider is the benefit of considering residual nodal disease rather than using a system which incorporates response information from the breast as well as the nodes?

A slightly more detailed discussion of the limitations of the data in the discussion section would be beneficial.

I would not entirely agree with their statement that “Little is known about the impact of survival outcomes of post-NAC….” (line 185). I think there is accumulating evidence (which this study adds to) that the prognostic effect of pCR varies across BC subtypes. I would consider that their conclusion from this data should be that it supports previous findings that pathological response has different prognostic implications across breast cancer subtypes, and specifically that pCR as defined by ypT0 ypN0 in ER+ HER2-ve (luminal disease) is not as predictive of outcomes as it is in TNBC.HER2+ disease, and I think that they should discuss this before making their very reasonable suggestion that the cut-off of 4 residually involved nodes be considered the high risk group in ER+ HER2-ve disease. I agree that a dichotomous pCR/no pCR classification is no longer sufficient and their data supports the argument that a reporting system incorporating this information should be routinely used following NAC in EBC to provide this level of granular detail.

A couple of minor points:

Table 4 contains too much detail and the text is very difficult to read.

Line 210: I’m not clear what “weak effective of the latter category” means – does it just mean that the number of patients in this group is very small?! Please clarify.

Author Response

We would like to thank reviewer #4 for the important comments and the essential revisions he/she suggested. Please find enclosed in the attached document our response to reviewer #4. 

Round 2

Reviewer 1 Report

It is a great pleasure to see significant improvement of the manuscript during
the review process. I appreciate and agree with responses from the authors. The paper is publishable in its current format. However I have two minor questions on the revised manuscript. It would be nice to hear from the authors about their responses:

1. The anti-HER2 information adds a lot to the audience to evaluate relevance to
the clinical practice. However in the modified portion of table 1, 60.8% (n=258
out of 424) of patients receive adjuvant anti-HER2. Comparing to 24.6% HER2
positive rate in the whole study population (the original table 1, 295 out of
1197), it raises a question what constitutes the denominator for patients to be
analyzed for adjuvant anti-HER2 therapy (who are those 424 patients?)

2. In the modified table 1 radiotherapy session: "sus-clavicular radiotherapy"
appears to be a French spelling for supra-clavicular radiotherapy. May consider
to revise to an English version. If possible, please consider to include internal
mammary chain radiotherapy in this table as this is another lymphatic drainage
target to be covered by regional nodal irradiation.

Reviewer 3 Report

I am very much convinced by the revised version of the manuscript. The paper has improved and should be published now.